

# An empirically-derived hydraulic head model controlling water storage and outflow over a decade in degraded permafrost rock slopes (Zugspitze, D/A)

Riccardo Scandroglio[1], Samuel Weber[2,3], Till Rehm[4], and Michael Krautblatter[1]

[1]Landslide Research Group, TUM School of Engineering and Design, Technical University of Munich, Munich, Germany
[2]WSL Institute for Snow and Avalanche Research SLF, Davos Dorf, Switzerland
[3]Climate Change, Extremes and Natural Hazards in Alpine Regions Research Center, CERC, Davos Dorf, Switzerland
[4]Environmental Research Station Schneefernerhaus, Zugspitze, Germany

**Correspondence:** Riccardo Scandroglio (r.scandroglio@tum.de)

## 1 Abstract

While recent permafrost degradation in Alpine peri- and paraglacial slopes has been documented in several studies, only restricted information is available on the respective hydrology. Water boosts permafrost degradation by advective heat transport and destabilizes periglacial mountain slopes. Even if multiple recent rock slope failures indicate the presence of water, only

5  a few studies provide evidence of water availability and related hydrostatic pressures at bigger depths, showing a significant research gap. This study combines a unique decennial data set of meteorological data, snowmelt modeling, and discharge measurements from two rock fractures in a tunnel located ≈ 55 m under the permafrost-affected N-S facing Zugspitze Ridge (2815-2962 m asl). To decipher the hydrological properties of fractures, we analyze inputs, i.e., snowmelt and rainfall, and outputs, i.e., discharge from fractures, baseflow, and no-flow events, detecting flow anomalies. For summer precipitation events,

10  we developed i) a uniform recession curve, ii) an empirical water storage model, and iii) an approximate hydraulic water pressure model according to Darcy's falling-head law. Extreme events with up to 800 l/d and 58 l/h are likely to fully saturate the observed fractures with corresponding hydraulic heads of up to 40 ± 10 m and to increase fracture interconnectivity. The average daily discharge during snowmelt, 10 l/h, can lead to hydraulic heads up to 27 ± 6 m. Water dynamics suggest hydraulic conductivities in the range of $10^{-4}$ m/s, with variations according to the fracture's saturation. E.g., no-flow and

15  baseflow events indicate unsaturated and partially saturated conditions. Here, we show an empirical fluid flow approximation model of hydrostatic pressure regimes in high-alpine deep-bedrock fractures. Pressures from water accumulation in bedrock reach levels that can weaken or even destabilize rock slopes. This process can easily outpace thermal conductive warming of active layers in the foreseeable future, provide positive feedback on water infiltration, and is crucial for the stability of the rapidly warming alpine permafrost environments.



## 2 Introduction

High mountain regions are "global water towers" (Viviroli et al., 2007), which massively sustain seasonal water availability for ∼ 1.9 billion people (Immerzeel et al., 2020), but knowledge of groundwater dynamics in high-alpine rock slopes is scarce. Most studies focus on sub-alpine watersheds with soil and vegetation covers and infer groundwater dynamics only through spring discharge (Hayashi, 2020) due to the lack of hydrological measurements at high elevations. Logistically challenging terrain, harsh meteorological conditions, and substantial pattern variability (Walvoord and Kurylyk, 2016; Arenson et al., 2022) are limiting factors in high alpine environments. So far, only a few studies have directly monitored groundwater in alpine hillslopes with deep wells in competent and fractured bedrock (Manning and Caine, 2007; Gabrielli et al., 2012). The models applied often assume that bedrock in alpine catchments behave like "Teflon basins" with flow only at the surface (Clow et al., 2003) unless additional sediment-rich surface layers are available.

Developments suggest that deep fractures are a crucial pathway for groundwater flow along the hillslope and that this can be highly dynamic (Banks et al., 2009). Steady baseflow during periods of little recharge indicates the relevance of aquifers (Hayashi, 2020), contributing 5–50% to adjacent lowlands aquifers (Markovich et al., 2019), seasonally redistributing water, and stabilizing catchment outflow (Cochand et al., 2019). Geological conditions mainly control groundwater in mountain bedrock and can be divided into two components. On the one hand, the main flow component is shallow and is topographically driven because of the higher weathering in the first meters (Clarke and Burbank, 2011; Welch and Allen, 2014). On the other hand, the deep bedrock flow is characterized by a complex and heterogeneous permeability reduction with depth (Manning and Caine, 2007), and it is controlled, among others, by fracture density, geometry, and connectivity, which can be highly complex and heterogeneous.

Permafrost is known to influence groundwater flow paths and storage (Woo, 2012) and allows the accumulation of perched water table above the frozen material (Krautblatter et al., 2013). Still, it remains unclear to what extent thawing mountain permafrost contributes to the water cycle, e.g., through groundwater storage (Walvoord and Kurylyk, 2016). Mountain permafrost hydrology studies mainly focus on unconsolidated sediments (Hayashi, 2020), since the majority of alpine springs discharge comes from talus, moraine, or rock glacier (Noetzli and Phillips, 2019; Jones et al., 2018; Arenson et al., 2022). Only recently, Ben-Asher et al. (2023) combined field measurements and numerical modeling to simulate hydrological fluxes on steep bedrock permafrost. However, much work is still to be done to understand bedrock hydrology in periglacial areas. From the slope stability point of view, permafrost bedrock failure results from many factors, e.g., joint sets geometry, presence of a fault zone, or glacier retreat (Haeberli et al., 1997; Hasler et al., 2012; Phillips et al., 2017). The presence of water in the scarp after rock slope failures, as well as the occurrence of high water availability prior to the failure, has been observed recently in various events (Stoffel and Huggel, 2012; Fischer et al., 2010; Walter et al., 2020; Kristensen et al., 2021), suggesting that high hydrostatic pressure contributes to failure. According to Montgomery et al. (2002), near-surface fractured bedrock can influence landslide triggering positively or negatively, depending on pressure head build-ups or storm runoff accommodation. In the rock-/ice-mechanical model by Krautblatter et al. (2013), hydrostatic pressures can lead to increased lateral shear stress on the rock mass, reduce frictional strength, and lower effective normal stress leading to a reduction in shear resistance of





rock-rock contacts. Active-layer deepening due to climate change increases the infiltration depth of water, generating higher
hydrostatic pressures (Haeberli and Gruber, 2009). Recent slope-stability models with and without permafrost focused on the
role of water in fractures, proving that it plays a crucial role in slope stability (Scandroglio et al., 2021; Magnin and Josnin,
2021). From the thermal point of view, water percolating in bedrock can quickly thaw fractures by advection and, therefore,
destabilize bigger rock masses than heat conduction (Haeberli et al., 1997; Gruber and Haeberli, 2007), but so far, direct field
evidence of water availability and consequent sudden thermal disturbance at depth is only available for one site (Gemsstock
(CH), Phillips et al., 2016).

Different approaches to detect and quantify water presence range from multi-method geophysics in permafrost (Hauck et al.,
2011; Wagner et al., 2019; Pavoni et al., 2023) and in unfrozen karst (Watlet et al., 2018), over piezometric measurements in
frozen cracks (Draebing et al., 2017) and rock glacier (Phillips et al., 2023; Bast et al., 2024), to surface lysimeters (Courtin
and Bliss, 1971; Rist and Phillips, 2005). So far, all methods have shown substantial limitations in bedrock, and water pressure
at deep depths has not been directly measured yet.

Despite the recent increasing importance and interest in this research field, we still lack measures and understanding of how
deep bedrock groundwater ($> 10\,m$) couples to superficial water availability on a short-term (days) and long-term (seasonal)
scale in alpine environments. Inferring water accumulations and their destabilizing potential remains a core challenge that
requires further research. In this study, we combine a decade of underground discharge measurements, weather data, and snow
simulations to derive an empirical model of water dynamics in the first decameters under the surface ($\approx 55\,m$) of high-alpine
low-porosity bedrock. We address the following questions:

i) How can we estimate fluid flow's spatial and temporal behavior in near permafrost bedrock fractures?

ii) How can we use fracture depletion to constrain fluid flow curves?

iii) How can we use outflow curves to constrain water storage?

iv) Can we generate a Darcy falling head model to mimic and generalize fluid flow in bedrock fractures?

## 3 Study site description and characterization

Measurements took place on Mount Zugspitze (2962 m asl, Fig. 1a), located in the Northern Calcareous Alps at the German-
Austrian border and visited by thousands of tourists daily. The study area is located southwest of the summit, on the east-west
oriented ridge between Zugspitze and Zugspitzeck. A disused tunnel for pedestrians runs from southwest to northeast under
this ridge (Fig. 1b), and it is accessible all year round thanks to its direct connection to the research station Scheefernerhaus
(UFS).

### 3.1 Climate and cryosphere

Long-term meteorological records by the German Meteorological Service (DWD) have existed since 1900 on the summit
(Fig. S1 in the supporting materials). The mean temperature in the last decade (2013-2022) was $-3.3°$C, which is $1.5°$C



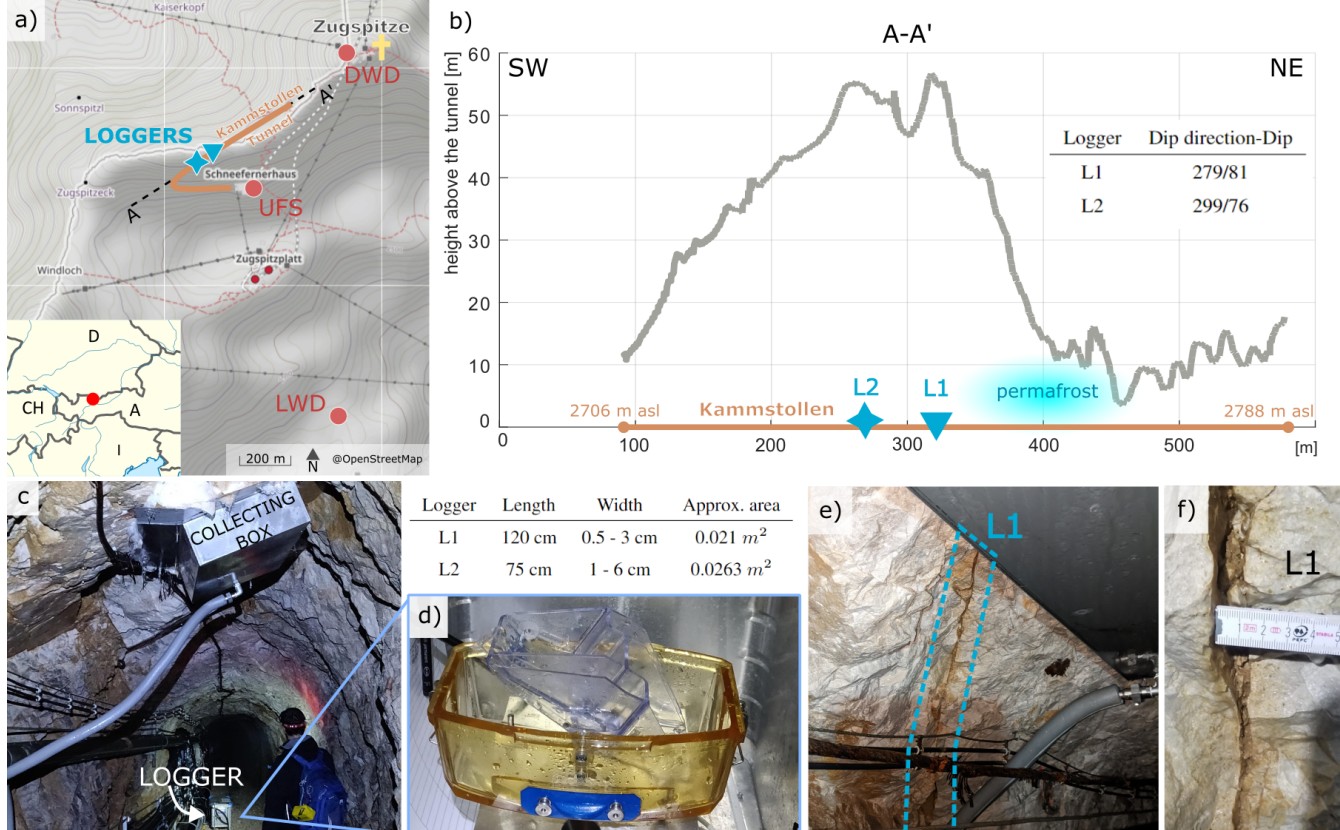

**Figure 1. Overview of the study site.** a) Zugspitze summit (yellow cross, 2962 m asl) with the location of the Kammstollen tunnel in orange. Blue signs represent the two flow loggers, and red circles the weather stations: DWD = German Meteorological Service, UFS = Environmental Research Station Schneefernerhaus, and LWD = Bavarian Avalanche Service. ©OpenStreetMap contributors 2023. Distributed under the Open Data Commons Open Database License (ODbL) v1.0. b) Section A-A' of the ridge along the tunnel showing the distance between the tunnel and the surface. The tunnel ascends from 2706 to 2788 m asl; the loggers are at approx. 2750m asl. The location of the loggers and the permafrost lens (Krautblatter et al., 2010) is shown. The table presents the orientation of the two fractures with loggers. c) The collecting box is installed on the ceiling and is connected to the logger with a pipe. d) Tipping gauge with resolution 0.1 L and table with measures of both fractures. The approximate fracture area covered by the sample is length by width. c) Fracture L1 and d) detail of L1 with scale.

warmer than the reference period 1961–1990, with 2022 being the warmest year ever recorded, with $-2.7°\,$C on average. The climate is influenced by the prominent elevation at the northern edge of the Alps and by multiple E-W oriented ridges, leading to mean annual precipitation of more than $2500\,$mm, with no recent changes compared to the reference period. 80% of precipitation is snowfall from autumn to late spring, most parts of the catchment are snow-free at the end of July, and heavy thunderstorms, as well as long-lasting, intense rainfall events, may occur during the summer season (Wetzel et al., 2022).



Mayer et al. (2021) documented the dramatic recession of the three glaciers at the Zugspitze: with no mass increases since the 80s, the glaciers are destined to disappear in the next decade. Permafrost occurrence and degradation are well documented and closely monitored at the summit (Gallemann et al., 2017) and in the tunnel (Krautblatter et al., 2010). The permanently frozen bedrock of the tunnel is located a few decameters from our study site. Geophysical measurements are conducted monthly to monitor the evolution of the permafrost lens and detect possible interaction with percolating water. An ice-controlled slope instability has been observed on the ridge, 400m from our site, and it was monitored recently by Mamot et al. (2021) with permafrost geophysical mapping and deformation monitoring.

## 3.2 Hydrology

The basin south of the summit, also known as Research Catchment Zugspitze (RCZ, 11.4 km$^2$), is one of the best instrumented high alpine catchments for monitoring hydrological processes thanks to a high-density sensors network over four elevation zones and hydrograph recording at the downstream Partnach spring (Wetzel, 2004; Weber et al., 2021). Hydrochemical investigations by Rappl et al. (2010) and Weishaupt (2021) using natural environmental tracers and electric conductivity provided the catchment borders and their hydrogeological characteristics, which prove the catchment to be a perfect natural lysimeter. They evidenced a karst water reservoir in the phreatic zone beneath the Zugspitze Plateau that can hold around half of the volume of summer precipitation. During the winter season, from the end of October to April, no karst system recharge occurs, and Partnach-Spring falls dry (Morche et al., 2008). Because of this dynamic, climate-change-induced variations in the snow cover will strongly affect water availability in the RCZ (Weber et al., 2016). Furthermore, Voigt et al. (2021) used relative gravity measurements to detect water storage variations in the RCZ with promising results. Still, the only known measurements of water discharge in shallow unsaturated bedrock were conducted for measuring persistent organic pollutants in shallow percolated water (Levy et al., 2017), using the same loggers as in this study.

## 3.3 Geology and fractures

The whole summit area is composed of Triassic Wetterstein limestone, with a thickness of about 600–800 m (Hornung and Haas, 2017). Ulrich and King (1993) report brecciated zones up to 1 m thick that dip steeply (60°–90°) in the directions of NW–ENE and can be intercalated with ice. A relevant fault zone can be found from above the UFS up to the summit, and karst dissolution is frequent, especially on the Plateau. The fractures in the tunnel were mapped in 2007 Krautblatter et al. (2010) and are here newly analyzed in Figure S2 of the supplementary material. Fractures with a dip of 80-90°are the majority (S1, n=41) but without a predominant direction (SD = 90°). Few other discontinuities are mapped, mainly with orientation 58/106°(S2, N=7) and 49/311°(S3, N=11). This mapping agrees only partially with the one from Mamot et al. (2021), which conducted scan lines and field mapping 400m NW from our site (Fig. S2e). In addition, punctual measurements on the two main fractures, where the water gauges are installed, were conducted by Georg Stockinger in 2023 and are presented in Figure 1b.





# 4 Methods and data

In this study, we present a modeling approach to estimate the hydraulic head (hereafter also called hydrostatic pressure), revealed from rain and snowmelt time series (system input) and measurements of water discharge from fractures in the tunnel (system output). Using this information, we decipher the fluid flow in deep bedrock fractures (system process).

## 4.1 System input: water from rain and snowmelt

Inputs to the model can be rainfall from the weather stations or snowmelt from the software Snowpack. Meteorological measurements are conducted at three locations, shown in Figure 1 and presented in Table 1.

    i) DWD - The German Meteorological Service records air temperatures every 10 minutes and the daily precipitation on the summit of the Zugspitze. Due to the exposed location of this weather station, results are often influenced by winds and northern weather.

    ii) UFS - The Environmental Research Station (2650 m a.s.l.) measures precipitation and air temperatures every 10 minutes together with the DWD. In the middle of the south slope, this location is protected from the northern winds but exposed to western atmospheric perturbations.

    iii) LWD - The Bavarian Avalanche Service runs an automated snow and meteo station on the Zugspitzplatt (Plateau, 2420 m asl). This station is in a central and flat position, protected from strong south, west, and north winds. Values are recorded with 10 minutes resolution.

| Station | Parameter | Resolution | Time analysed | Usage |
|---|---|---|---|---|
| DWD | TA, PSUM | yearly, monthly, (hourly, daily) | 1901-2023 | long-term trends |
| UFS | TA, PSUM | 10 Min. | 2000-2023 | rainfall modelling |
| LDW | HS, TA | 10 Min. | 2000-2023 | snow modelling |

**Table 1.** Metereological input used in this study. TA = Air temperature, PSUM = sum of precipitation, HS = snow height.

For the snowmelt, we use the one-dimensional open-source software Snowpack (SP) that models the evolution of the snow cover based on weather data. It simulates mass and energy exchanges taking place between the atmosphere, snow, and soil and reproduces the evolution of snow microstructure (Lehning et al., 1999). Simulations were conducted with data from LWD for every hydrological year separately, with 15-minute time steps. The provided inputs are incoming and outgoing shortwave radiation, snow depth (HS), relative humidity, air temperature (TA), sum of precipitation (PSUM), temperature of the snow surface, and wind speed/direction. The measured snow depth is a proxy for precipitation inputs to force the mass balance. Data



gaps are interpolated with the integrated pre-processing library MeteoIO. Ground temperature is set constant at 0°C and albedo
is estimated from incoming and reflected shortwave radiation. Boundary conditions for snow melting are adapted each year to
fit the melting phase best, but due to model limitations, discrepancies between modeled and measured snow heights are still
possible. The most important output is the amount of meltwater $[kg/m^2]$ that leaves the snow cover in liquid form and reaches
the ground.

## 4.2   System output: water discharge from fractures

Two water collecting systems were installed in 2010 in the central part of the pedestrian tunnel to collect water that leaks from
two independent fractures (POPALP Report, 2011; Levy et al., 2017). Each system is composed of one collecting box installed
on the ceiling of the tunnel (Fig. S3 in the supplementary material), a connection pipe, one tipping bucket with a reed sensor
for measuring the discharge, and one logger for data recording. The boxes are located in the unfrozen area, as confirmed by
electrical resistivity tomography (Krautblatter et al., 2010), and their vertical distance to the surface obtained with a DTM
is $\approx 55\,\mathrm{m}$. The fractures corresponding to each logger and their measures are shown in Figure S3 (see supporting material).
Discharge data since 2013 are available thanks to the UFS staff; new loggers were installed in 2020.

## 4.3   System process: fluid flow in deep-bedrock fractures

The system inputs and outputs are united for analyzing and modeling fluid flow. We first perform a systematic analysis to
characterize it, then model water flow and accumulation in fractures based on (i) recession curves analysis, (ii) detection of
flow anomalies, and (iii) assessing of fracture saturation. Finally, we suggest an approach to quantify the hydraulic head.

### 4.3.1   Systematic analysis

Discharge measurements, meteorological data, and *SP* results were united in Matlab at hourly and daily resolution. Snowmelt is
analyzed seasonally and daily, while rainfall is only on an event basis. Snowmelt produces uninterrupted flow for many weeks
up to months, making it hard to define single events. Meanwhile, rainfall is naturally event-based, and often dry periods clearly
mark the limits. Mixed events are excluded, while rain-on-snow cases in spring are included. An output flow event starts with
a sudden increase of the discharge, independent of the starting value, and ends when the flow returns to a value smaller than a
threshold (typically 1 l/h). Baseflow is a constant or very slowly decreasing discharge of small magnitude (typically $< 1$ l/h)
that happens after a flow event and can last up to some weeks, even without further input. By convention, multiple flow events
are classified as one if precipitation interruptions are shorter than 24 hours and if the resulting hydrograph at the gauges does not
reach baseflow status between the two rain events. Input-output anomalies in the flow are detected and analyzed. One example
is no-flow events when rain generates no relevant flow in the fracture. After a manual pre-selection, timing and quantities are
analyzed automatically with a dedicated Matlab function where parameters are provided by the user. They are variable from
case to case due to the heterogeneity of the events. All selected events are listed in Figure S1 of the supplementary material,



together with the input-output graphics for each event in Figure S8 and S9. Some analyses are limited to fracture L1 since L2 recorded less events.

### 4.3.2 Model fluid flow and water accumulation in fractures

We only model rain events because they can be clearly separated from each other thanks to the dry phases in between. Since input and output quantities are all directly measured, the resulting empirical model is certain. On the contrary, snowmelt is the output of SP modeling; therefore, quantities there must be considered a likelihood and not a precise measure. Additionally, during snowmelt, water flows uninterrupted for days to weeks, making it harder to detect single events.

**Recession-curve analysis**

The recession-curve analysis represents a basic hydrogeological research tool, used for over a century but still up-to-date, with modifications and expansions. Recession curves can be fitted to drainage data to fully reproduce the runoff with empirical coefficients. This provides information on the flow characteristics and on the attributes of the aquifer, e.g., estimation of karstification degree or groundwater sensitivity to pollution (Malík and Vojtková, 2012; Kirchner, 2009). Boussinesq (1877) was the first to describe aquifer drainage and spring discharge through a porous medium using a diffusion equation. Using

simplifying assumptions, he obtained the approximate analytical solution described by exponential Equation 1, where $Q_0$ is the initial discharge, $Q_t$ is the discharge at time $t$, and $\alpha$ is the recession coefficient, an intrinsic aquifer parameter.

$$Q_t = Q_0 e^{-\alpha t} \tag{1}$$

Maillet (1905) also obtained similar results for reservoir emptying through a porous plug: knowing initial discharge values and recession coefficients, the entire runoff curve can be described. Hydrogeologists widely use Equation 1 due to its simplicity and

linearisation in logarithmical plots, but other recession equations are also available to better fit various shapes of hydrograms. For example, linear equations are used for fast-flow components typical from karstic channels (Malík and Vojtková, 2012). Complex aquifers with mixed flow regimes (e.g., with karst and multiple conduits) require a combination of different equations to reproduce their groundwater circulation. One or more laminar and turbulent sub-regimes may exist in one aquifer, and the total discharge can be described by the superposition of all the flow equations.

We took all the events and extracted the recession part of the discharge curve. If one event shows multiple peaks, the event has been divided into sub-events, and one curve has been assigned to each peak. All the curves are first plotted starting at time $t = 0$ (Fig. S4a in the supplementary material). Then, recession curves are analyzed from the biggest to the smallest and each curve is shifted in time so that its starting value fits the same value of the bigger event (Fig. S4b). We computed the mean of these shifted curves and tested different fittings to the mean curve, according to the procedure explained in Malík (2015). A

function that is the sum of several exponential segments (Eq. 2) is used to represent the complete recession hydrograph of a karst spring.

$$Q_t = \sum_{i=1}^{n} Q_{0i} e^{-a_i t} \tag{2}$$

 

**Flow anomalies detection**

During data collection, unusual situations were observed where the presence or absence of water in the tunnel could not be
directly explained. To understand these cases, we analyze all possible situations in Figure 2: after separating between snow-
covered and snow-free periods, we considered snowmelt (SM), precipitation (PSUM) and air temperature (TA), listing all
possible combinations. Finally, we highlighted unexpected flow behaviors in orange and looked for reasonable explanations:
all these situations could be explained by water accumulation in the bedrock layer above the tunnel.  Possible accumulation

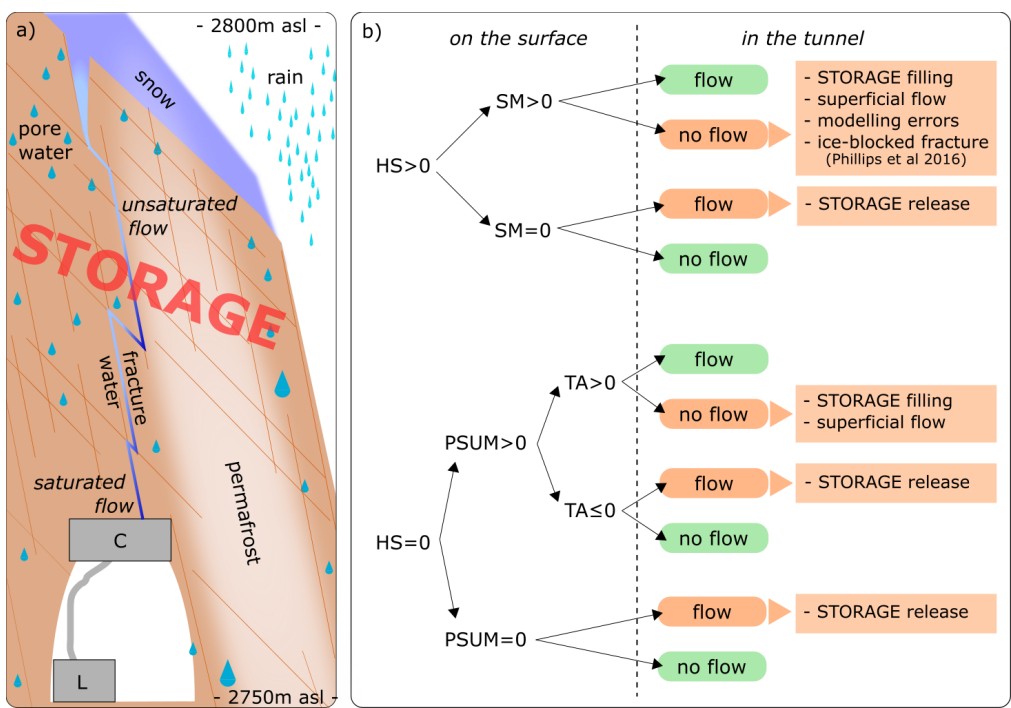

**Figure 2. Flow anomalies detection.** a) Simplified model representing all the components of hydrological significance. Storage includes
fractures, karst voids, and pores, i.e., all locations where water can accumulate. In the tunnel, we find the collecting box (C) and the logger (L).
b) Detection of unexpected water behavior in the tunnel; the upper case represents snow-covered periods, and the lower one represents snow-
free periods. Abbreviations: SM = snowmelt, PSUM = sum of precipitation, TA = air temperature. Green boxes show standard situations,
while the red boxes highlight situations requiring water accumulation within the rock mass.

places within the bedrock, hereafter abbreviated as *storage*, are (i) karst voids, (ii) pores, and (iii) fractures. i) The presence
of karst voids in the region of the Zugspitze Plateau is well documented (Wetzel, 2004), but their presence in the area above
the tunnel has not been proven yet. Karst would also strongly influence the flow behavior, and it is, therefore, unlikely. ii)
An average rock matrix permeability of 5 $\mu D$ (Krautblatter, 2009) limits the amount of water that can flow through the rock.
iii) Fractures, due to their filling with fine material, are preferred flow paths and can temporarily store water in accumulation
periods. Therefore, we focus our model on this last component.



**Fracture saturation and storage model**

Here, we present an empirical model that explains flow anomalies with fracture saturation and storage variations (Fig. 3). The model is composed of four stages: from unsaturated (S1) to partially saturated (S2a/b) to fully saturated (S3). Repeated alternation of steps 2b and 3 is possible when multiple precipitation events happen close enough in time. During dry periods, S1 resumes again until the next precipitation. Flow events can only happen if fractures are fully saturated, while baseflow is the outcome of partially saturated fractures. The main outputs of the model are i) a qualitative forecast of the saturation level in the fracture and ii) a quantitative estimate of the amount of water present in the fractures at each time step.

For ii), we first calculated the cumulative quantities at the end of the event: the discharge ($Q$) in [liters] and the precipitation ($PSUM$) in [mm] or [l/m$^2$]. While the first is exactly known for every event, the exact amount of precipitation infiltrating the fracture is unknown. Therefore, for each event, we use Equation 3 to calculate the ratio $R_{end}$ that expresses the surface required to obtain the measured flow in the tunnel.

$$R_{end} = \sum_{t=0}^{end} Q \Big/ \sum_{t=0}^{end} PSUM \tag{3}$$

Since we excluded superficial runoff and evaporation, this value is only a minimum estimation. $R_{end}$ is then used in Equation 4 to compute the storage fill level $FL_t$ at each time step $t$, that is, the difference between cumulative input and output.

$$FL_t = \sum_0^t PSUM * R_{end} - \sum_0^t Q \tag{4}$$

### 4.3.3 Conversion in water column

Finally, we estimate the resulting hydraulic head using Darcy's Law (Eq. 5), the basic equation that describes fluid flow through saturated porous media.

$$q = -K\frac{\Delta h}{\Delta l} \tag{5}$$

Here, y$q$ is the specific discharge $[LT^{-1}]$, $K$ is the hydraulic conductivity $[LT^{-1}]$, $h$ is the head $[L]$, and $l$ is the travel distance $[L]$ (Zha et al., 2019). According to Bernoulli's equation (Eq. 6), the total hydraulic head $h_t$ can be composed of elevation head $h_z$, pressure head $h_p$, and velocity head $h_v$. In this case, $h_v$ can be neglected due to extremely low velocities in porous media, and $h_p$ should be overall the same. Therefore, the elevation head $h_z$ is the dominant component and it drives water flow.

$$h_t = h_z + h_v + h_p \approx h_z \tag{6}$$

To constrain the hydraulic head, it is reasonable to compare our case to a falling-head test with a Darcy cylinder as in Figure 4a, where the only constant head is at the discharge point (for example in Allan Freeze and Cherry, 1979). Suppose the fracture is simplified with a cylinder having a constant diameter over the whole length ($a = A$, in Fig. 4b). In that case, $L$ is the "effective path" that offers hydraulic resistance, and therefore, a hydraulic head can build up above it. In Figure 4a, the discharge according





**Figure 3. Fracture saturation model.** Upper part: four phases of our model. Lower part: exemplary summer rainfall event. Precipitation (blue bars) from UFS, discharge (green line) from Logger 1, and the corresponding fill level of the storage in the fracture from Eq. 4 (red line). Point A) represents the maximum storage in the fracture system, and point B) represents the storage at maximum discharge. Table with steps of the fracture saturation model as a consequence of input and output.





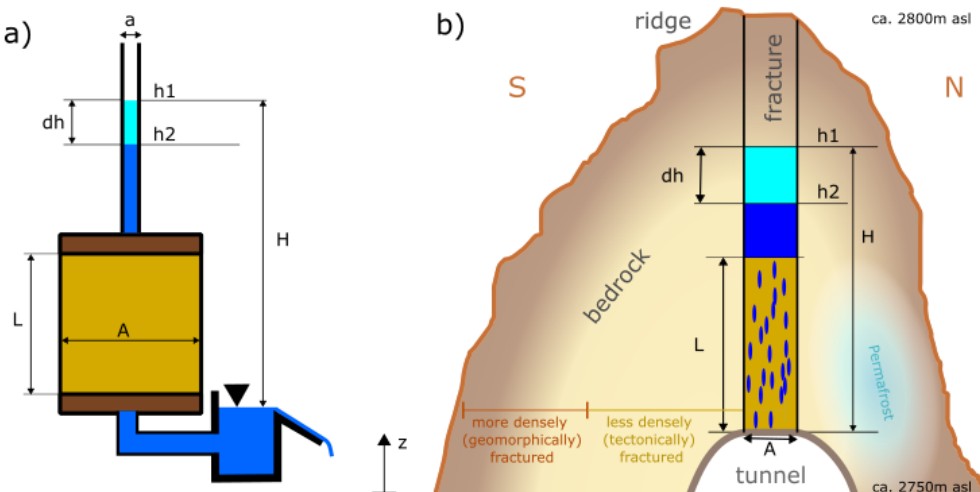

**Figure 4.** Application if Darcy's falling head test. a) Darcy cylinder with falling head in the pipe above and constant head at the outflow. b) Illustrative S-N transect of the fracture for water flow with falling head. Bedrock deeper than 8 meters is expected to be less densely fractured (Clarke and Burbank, 2011).

to Darcy is $Q = qA = -K \cdot A \cdot H/L$ and it must be equal to the change in hydraulic head, i.e., the discharge rate above the effective path $Q = a \cdot dh/dt$.

$$a\frac{dh}{dt} = -KA\frac{h}{L} \tag{7}$$

Since here $a = A$ we can simplify, rearrange the terms and integrate the equation, obtaining:

$$\int_{h_1}^{h_2} \frac{dh}{h} = -\frac{K}{L}\int_{t_1}^{t_2} dt \qquad\qquad ln(h_2) - ln(h_1) = -\frac{K}{L}(t_2 - t_1) \tag{8}$$

If we solve for the maximum event length, we suppose $t_1 = 0$ and $t_2 = t_{max}$, obtained from the recession curve. Considering the baseflow at the end $h_2 = 0.1\,\mathrm{m}$, we can compute $h_1$, the hydraulic head at the beginning.

$$h_1 = 0.1 \cdot e^{(K \cdot t_2/L)} \tag{9}$$

## 5  Results and data interpretation

Water flow is categorized into spring snowmelt and summer rainfall, evaluated separately in this chapter's first two sections. The third section focuses on no-flow events, baseflow, and extreme events, while the following sections compute the recession curve and analyze the storage. After defining some premises, the last section estimates the hydraulic heads, linking them with discharge values.



## 5.1 Snowmelt induced discharge

Daily (a) and hourly (b) snowmelt in spring 2023 is shown in Figure 5, while all seasons are available at daily resolution in Figure S5 of the supplementary material. Figure 6 presents snowmelt statistics from 2013 to 2023.

**Seasonal and daily analysis -** Snow melting generally starts at the end of April and lasts until the end of June (Fig. 5a).
Small events are also possible in summer or autumn, mostly with negligible snow. Daily values flow rates present good temporal agreement between measurements and model for the starting and the pauses, but discharge stops earlier than the model (e.g., in 2019, 2021, and 2022). The daily modeled melting rates vary strongly across the season: values increase with time, reaching the maximum at the end. Snowmelt starts in the second half of April for the SP model and for L1, while L2 starts in the second half of May (Fig. 6a). SP and L1 differ only by one day, while L2 has a median delay of 28 days from SP (Fig. 6b).
Yearly peak snowmelt is concentrated at the end of June for SP, while both loggers reach their peak about two weeks before or earlier (Fig. 6c). Still, 5 cases show a very good fitting in time. The yearly peaks reach 80 mm/d for SP and 840 l/d for L1, but no correlation between variables is evident (Fig. 6d). The stronger variations of L1 and the fact that, on average, its flow is five times higher than SP can be explained by increased fracture interconnectivity for periods of high flow, so more surface contributes to these events.

**Hourly analysis -** Figure 5b highlights some disagreement between measured and modeled values at hourly resolution due to water travel time, storage effects, and possibly model limitations. According to the SP model, snowmelt occurs only during daily hours, from 4 to 21 o'clock, with a maximum at 13 o'clock and no flow during the night (Fig. S6a in the supplementary material). As expected, melting hours per day increased towards the end of June (Fig. S6b in the supplementary material): this explains the maximum daily melting rates of that period. On the contrary, water flows continuously in the tunnel with daily
cycles that vary in intensity and timing over the season, which can be divided into two phases (Fig. 5b). The *main flow* during night hours, marked in blue, and the *secondary flow* during daily hours, marked in red. The superimposition of hydrographs can explain these two flows: water is coming from at least two paths that have different lengths, fracture apertures, and/or filling compositions. In fracture L1, the end of an event corresponds to the start of the next. New events mostly start around 12 and have a maximum at 22 o'clock (Fig. S6c). The delay between the beginning of SP melting and water flow in the tunnel is 13 h,
while the delay of the peaks is only 11 h (Fig. 6e), but both show substantial variability. Here, we cannot exclude that the actual delay is a multiple of the calculated one. To verify this, we plotted the correlation between SP and L1 for daily values over 5 years (Fig. 6f; for each year separately Fig. S7 in the supplementary material). A maximum correlation of $> 0.7$ is reached at 0 and 1 day, therefore, the delay of peaks can be 11 h or 11+24 h. Variable delays can be explained by changes in the fracture saturation level or the interconnectivity: with high saturation, the hydraulic conductivity increases, and water flows faster. The
daily maximum flow rate for SP is 4.7 mm/h on average, with a maximum at 9 mm/l, while the values of L1 are 10 l/h on average, with a maximum at 58 l/h (Fig. 6g and i). No correlation is evident here. The daily cumulative flow of SP does not fit the hourly maximum flow rates linearly (Fig. 6h), while the daily total flow of L1 is linearly correlated to maximum flow rates (Fig. 6i). This is because the quantity of water that can be released by 1 $m^2$ of snow during one day is physically limited, but with higher discharges, more fractures are connected.

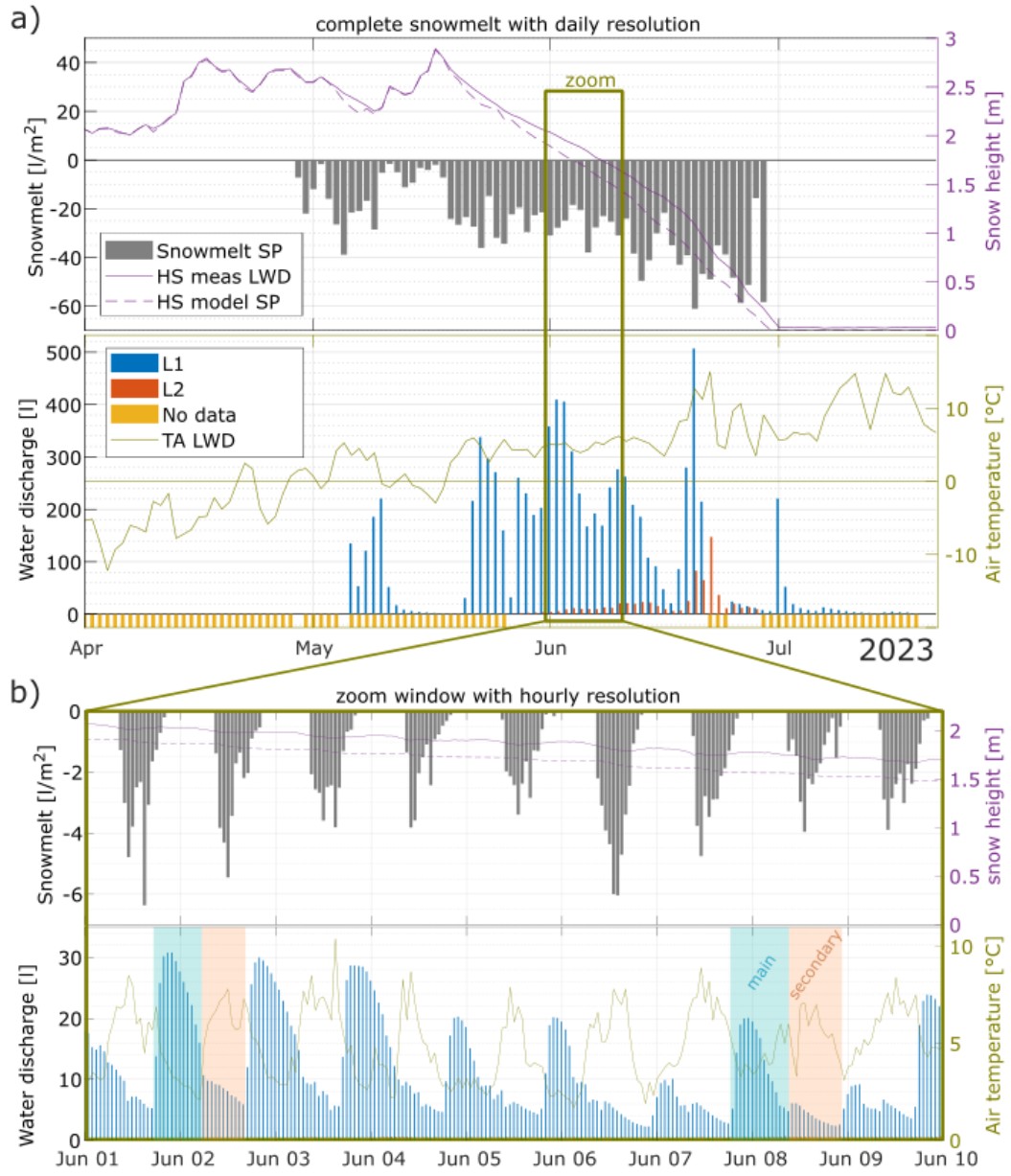

**Figure 5.** Example of modeled snowmelt rates and measured water discharge for summer 2023, daily rates in subfigure a) and hourly rates in subfigure b). The top panel of each subfigure shows the water input into the rock, i.e., melting water and snow height. The bottom panels show the water output, i.e., the discharge measured in gauges L1 and L2 (almost absent in this case), and air temperature. In the bottom panel of b), two different phases of water flow are highlighted: the main flow in blue and the secondary flow in red.



**Figure 6. Snowmelt statistics.** a) Violin plot of the flow start. The white points represent the median, and the darker areas represent the 25th and 75th percentiles. b) Delay between the Snowpack (SP) model and fluid flow loggers. c) Time of maximum flow for SP compared to L1 and L2. The line represents the same day. d) Maximum daily discharge for each year for L1, L2 and *Snowpack*, with probability distribution. e) Delay between SP and gauge L1 for flow begin and maximum discharge. f) Daily correlation between SP and L1, with the peak at 0 and -1 days. g) Maximum hourly flow for SP and L1 for each day. h) Maximum daily melting rate and correlated daily cumulative melting for SP. Excluded exceptions mostly represent rain-on-snow events. i) Maximum daily fracture flow and correlated daily cumulative flow for L1.

## 5.2   Rainfall induced discharge

This analysis is based on distinct rain events manually selected with high heterogeneity of quantities and duration. L1 recorded 23 rain events in the last 10 years, while it was possible to detect a relevant flow in only half of the cases for L2 (Tab. S1,

supplementary material). All possible flow trajectories after a rain event are explained in Figure 7. The output discharge from L1 and L2 can differ even with the same input, e.g., in Event 2 (yellow area) a flow is recorded only in L2, indicating different

saturation levels in the two fractures. For Event 3, on the contrary, discharge is recorded in both loggers, therefore both fractures are fully saturated.

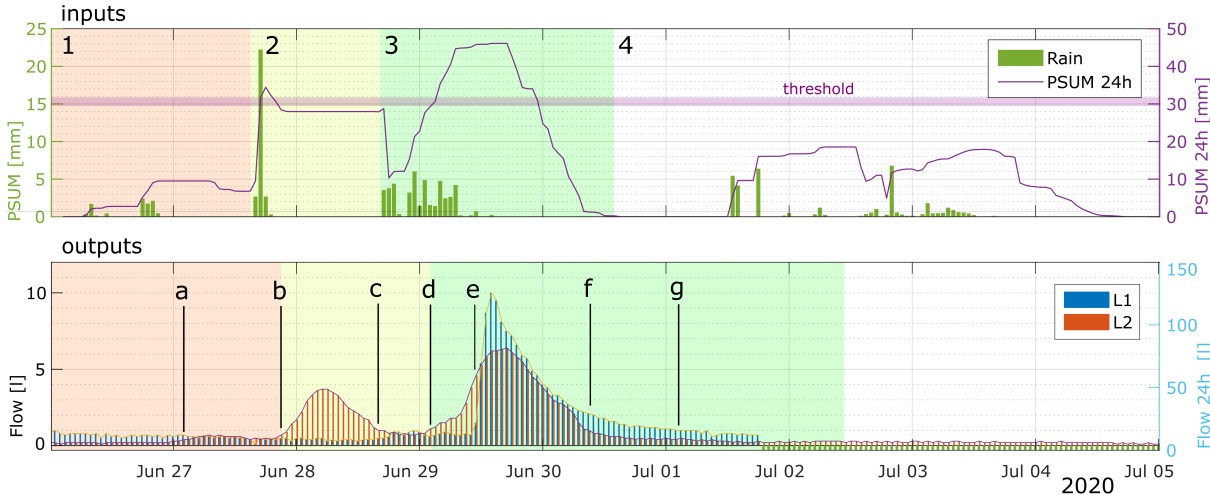

**Figure 7.** Example of a period with multiple summer precipitation events (i.e., inputs) and the corresponding flow in the fractures (i.e., outputs). Abbreviations: PSUM - sum of precipitation, PSUM-24h - sum of precipitation in the last 24 hours. The upper panel shows the inputs: 1) low-intensity short-duration event with PSUM-24 smaller than the threshold ($\approx$ 30 mm/24 h), 2) high intensity-short duration event around the threshold, 3) low-intensity long-duration events with PSUM-24 above the threshold and 4) again like 1). The lower panel shows the hourly outputs: L1 in blue and L2 in red. a) L1: no changes, L2: increase of baseflow, b) L2: begin of event, c) L1: increase of baseflow, d) L2: begin of second event, e) L1: begin first event, f) L2: end of the event, only baseflow, g) L1: inflection point, end of the event, only baseflow.

Results from the 23 events show that the duration of flow events in the tunnel is mostly longer than precipitation duration (Fig. 8a), but the correlation between duration in L1 and L2 is very high (Fig. S10 of the supplementary material). Even short rain events cause flow events that last longer than two days. There is a sudden increase in flow duration for longer precipitation

($>$ 80 mm), which could signify higher fracture interconnectivity. Flow in the gauge L1 can last up to 7 times longer than the precipitation (Fig. 8f). The delay from precipitation start to flow in the tunnel is 31 h for both loggers (Fig. 8b), but this time can vary $\pm$ 10 h according to the amount of precipitation occurring in the previous 3 days (Fig. 8c). This value is a good proxy for the pre-saturation level of the fracture: the more rain before the event, the higher the fracture's saturation, so the faster the water flows. Maximum hourly flow rates in L1 (3.9 l/h) are smaller than precipitation (8 mm/h) due to slow percolation

smoothing the peaks (Fig. 8d), but if we consider the 24h-sums, the discharge in the tunnel becomes proportionally bigger than the precipitation (Fig. 8f and Fig. S10). Total quantities confirm this: more than one liter of water reaches the tunnel for each mm of rain falling on the surface: 2.3 l/mm of rain in L1, 1 l/mm for L2 (Fig. 8e). These values, in m$^2$, express the minimum



**Figure 8. Analysis of rain events.** a) Correlation for event duration: PSUM compared to flow in L1 and L2. b) Violin plot of the delay between precipitation and flow in L1 and L2. c) Relation between precipitation in the three days before the event and delay for L1. d) Violin plot with maximum hourly precipitation and flow rate. e) The main graph shows the correlation analysis for the total quantities, L1 or L2 vs. PSUM. The small graph shows the rations L1/PSUM and L2/PSUM for total quantities. f) Ratios L1/PSUM for different quantities: event total duration, maximum flow rate, and 24h-cumulative maximum flow.

catchment size for the fractures: considering losses by evaporation and superficial runoff, we can suppose that, in reality, bigger areas contribute to each fracture. No events are recorded in the tunnel with less than ≈ 30 mm precipitation (Fig. 8e).

## 5.3 No-flow, baseflow and extreme events

**No-flow -** Some rain events do not produce a relevant water flow in the tunnel, and they are hereafter defined as no-flow events. We selected 49 summer rain events to investigate this anomaly (Tab. S2, supplementary material). No-flow events last up to 36 h





and have a maximum precipitation of 34 mm. The median of the 6h-sum of precipitation is very similar to the total precipitation, meaning that the precipitation is mostly concentrated in a few hours. Total precipitation and peak intensity are poorly correlated

with duration (Fig 9b), but total precipitation is related to the peak intensity (Fig 9c), confirming the predominance of short high-intensity rainfalls. No events show long duration and low intensity, indicating that rainfall generating no-flow events is mainly short or has high-intensity, likely thunderstorms.

**Baseflow -** The importance of baseflow is well highlighted in Figure 7, where baseflow precedes and forecasts the arrival of an event in the tunnel. Baseflow is also fundamental in our model (Fig. 3): during S2b, baseflow is explained by a partially

saturated fracture. During dry summer periods, baseflow reduces to zero if two precipitation events are far enough from each other, as in Sweetenham et al. (2017). In this case, we suppose the fracture to be unsaturated.

**Extreme events -** The logger recorded two special cases with extreme discharges. The first is the snowmelt on 10th-12th June 2019, which happened after a record snow depth of more than 6 m in May and in connection with a sudden increase in air temperature up to +10°C. This caused extreme tunnel discharges up to 800-750 l/d for three consecutive days. The second

extreme event is the rainfall on 16th-18th July 2021, with peak intensities of 160 mm/ 24 h and 20 mm/h, preceded by multiple smaller rain events in the five days before, which pre-saturated the fractures. This event was forecasted by the public warning service and generated floods in the valley. In the tunnel, discharges reached values above 800 l/d and hourly values up to 55 l/h, the maximum in the last 10 years. In this case, the delay between peak precipitation and peak flow is only 3 hours.

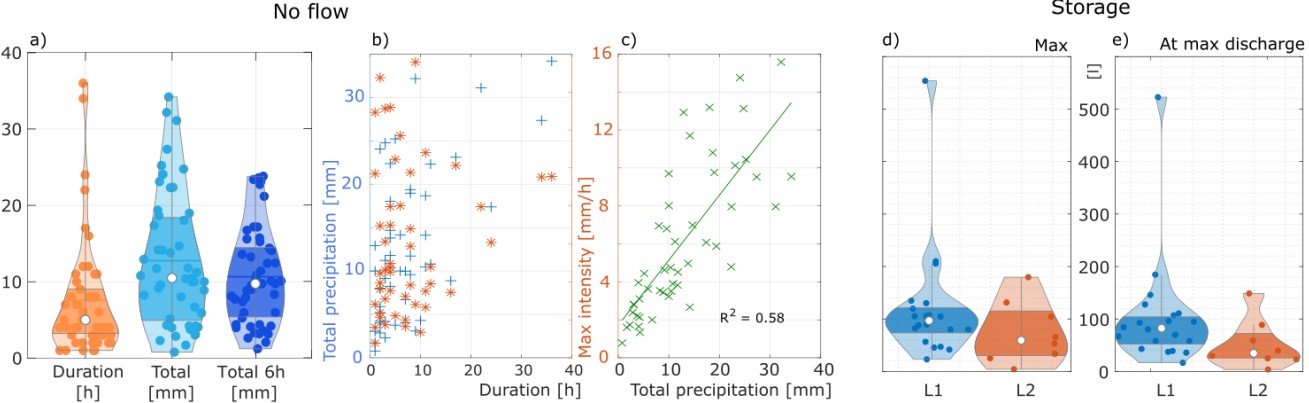

**Figure 9. Left: analysis of no-flow events** A) Statistics and spread for 49 events. b) Correlation between duration and total precipitation (blue crosses) or maximum intensity. c) Correlation between total precipitation and maximum intensity. **Right: analysis of storage.** d) Maximum storage and e) storage at maximum discharge.



## 5.4 Recession curve fitting

Comparing all the L1 discharge curves normalized from 0 to 1 in time and quantity shows that they follow a similar pattern (Fig. S11 and S12 of the supplementary material). On average, the discharge peaks after the first 10% of the time. Then, within only 30% of the time, it decreases back to 30%, and the remaining 60% of the time is required to return to baseflow.

We apply the recession analysis to the events from L1, as shown in Figure S4 in the supporting material. The best fitting is obtained with two exponential components, i.e., Equation 2 with n = 2, that reduces the error to an acceptable level (Table 2).

According to the classification suggested by Malík and Vojtková (2012), "the combination of two or more sub-regimes with merely laminar flow characterized by different discharge coefficients and higher values of $\alpha_x$." describes a flow happening in an "aquifer with irregularly developed fissure network, with the majority of open macrofissures, and with the possible presence of karst conduits of limited extent".

**Table 2.** Parameters of the mean recession curve fitting.

| Fitting | Flow component | $\alpha_x$ (or $\beta_x$) [1/h] | $Q0_x$ [l/h] | Error[1] |
|---------|----------------|-------------------------------|--------------|----------|
| Linear | fast-flow / turbulent | $\beta_1 = 0.023$ | $Q0_1 = 13.6$ | 4103 l |
| Exponential 1 | slow-flow / laminar | $\alpha_1 = 0.109$ | $Q0_1 = 41.9$ | 86 l |
| Exponential 2 | slow-flow / laminar | $\alpha_1 = 0.015 / \alpha_2 = 0.123$ | $Q0_1 = 1.8 / Q0_1 = 41.4$ | 17 l |

## 5.5 Empirical storage modeling

As next step, we use Equation 4 to compute the fill level $FL$ of the storage at each time step. The supplementary material presents the resulting storage curves for all events in Figure S13 and S14. The maximum water distributed in the fracture system is on average 97 l for L1 and 59 l for L2 (Fig. 9d). Since the timing of the maximum storage does not fit with the maximum discharge, we suppose that water reaches the maximum pressure when the maximum discharge in the tunnel is recorded (see points A and B in Fig. 3) and that all the water in the fracture is concentrated in one point, the bottleneck of the fracture. At 345 maximum discharge, the average storage reduces by 15-20%, but the highest value still reaches 520 l in July 2021 (Fig. 9e).

## 5.6 Estimating hydrostatic pressures from discharge measurements

With the information obtained so far and applying Equation 9, we constrain a realistic value range for $h_1$, the maximum hydraulic head at the beginning of a discharge event. We apply the following premises.

i) An absolute boundary condition comes from the height of the crestline surface above the tunnel: $h_1 < 55\ m$.

   ii) General bedrock porosity, including matrix and fractures, is estimated at $2.5 \pm 1.5$ % (Krautblatter, 2009), which gives an average storage capacity of $25 \pm 15$ l for each $m^3$ of rock. Considering the extreme event of July 2021, 520l must be





stored above the tunnel, resulting in $13 < h_1 < 52$ m. These numbers might be smaller, as we know that on the surface, porosity increases, and more fractures are present.

iii) The length of the *effective path*, $L$, must be realistic: $5 < L < 20\ m$. We fix one value for the whole process.

    iv) $t_2$ is the theoretical length of the maximum event and can be obtained from the recession curve analysis: $t_2 = 200$ h.

    v) A plausible range of $k$ is obtained from literature and from the recorded events. According to Allan Freeze and Cherry (1979), hydraulic conductivity $K$ in karst limestone can vary between $10^{-6}$ and $10^{-2}$ m/s. The travel time of water is known, i.e., start delay, and the flowing path is supposed to be the same as the distance surface-tunnel. The resulting

estimated field velocities are $\approx 5 \cdot 10^{-4}$ m/s.

Given these boundary conditions and applying Equation 9 to compute $h_1$, a plausible range for couples of $K$ and $L$ values are obtained (Fig. 10a and Fig. S3 of the supplementary material). These couples are then validated for different event lengths ($75 < t_2 < 200$ h) in Figure 10b. A realistic value of $K$ appears to be between $5 \cdot 10^{-5}$ and $1.5 \cdot 10^{-4}$ m/s, we choose the median $1 \cdot 10^{-4}$ m/s, which requires an effective length $L$ between 11.5 and 12.5 m to produce a max hydraulic pressure

$h1$ between 32 and 52 m. The validation of these results for shorter event duration confirms their feasibility. Each discharge requires a specific time to return to baseflow after an event (Fig. S4b, supplementary material). This is here defined as *time to 0* flow (=$tt0$), and can be obtained for any given discharge Q with Equation 2 and the parameters in Table 2. The resulting values of $tt0$ (Fig. 10c) are used in Equation 9 to compute $h1$ (Fig. 10d). This way, we connect discharges from the logger and hydraulic heads in the fracture. The fitting logarithmic curve for Q > 1 l/h produces $R^2$ = 0.99. Considering the average

maximum discharge from the rain events, Q $\approx$ 4 l/h, a hydraulic head of 20 m $\pm$ 4 m can be obtained. For snowmelt, we have an average daily Q $\approx$ 10 l/h that can generate a hydraulic head of 27 m $\pm$ 6 m. Extreme snowmelt in June 2019 could generate a hydraulic head of 40 m $\pm$ 10 m in the fracture. Similar values could have been reached after the intense rainfall in July 2021.

## 6   Discussion

This article provides valuable insight into the dynamic of water flow, reservoir effects, saturation levels, and pressure effects in

recently degraded permafrost fractures. It introduces an empirical method for quantifying hydrostatic pressures generated by snowmelt and rain infiltration. Differently from previous studies (Scandroglio et al., 2021; Magnin and Josnin, 2021), here, for the first time, we quantify pressures not based on modeling results but on a decade of underground discharge measures in high alpine fractures. The robustness of these flow data is the strength of this study.

### 6.1   Snowmelt- and rain-driven water flow dynamics in deep fractures as system input

Snowmelt modeled average and maximum daily infiltration rates are $\approx$ 30 mm/d and 80 mm/d, respectively, and are similar to those measured by Rist and Phillips (2005). The software SP can reproduce the timing of extreme melting events but shows disagreement at the end of the melting phase, due to the poor performance of the model or to the different locations of the snow





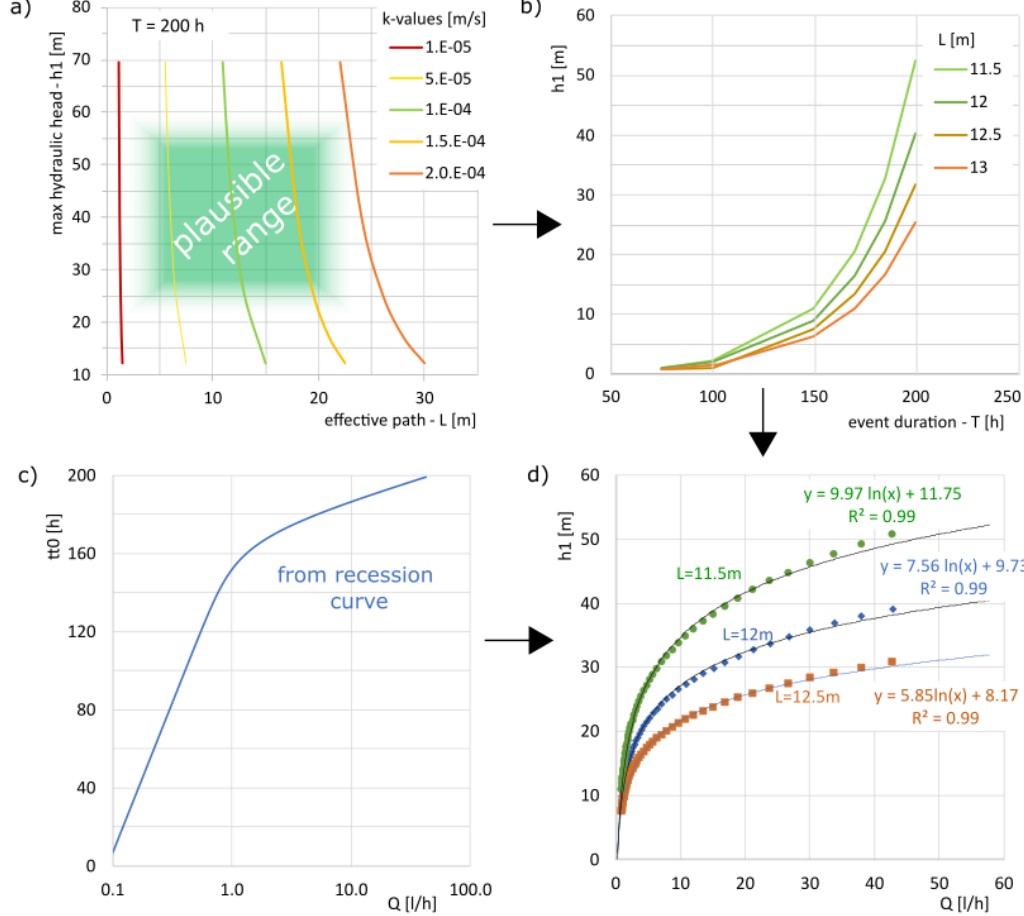

**Figure 10. Approximative hydraulic head model.** a) Hydraulic head resulting from different effective path (L) and hydraulic conductivity (K). The area in green highlights physically realistic results. b) Validation of the results by changing the event duration. c) *Time to 0 flow* (*tt0*) for each discharge $Q$. d) Hydraulic head $H1$ computed using discharge from fracture L1. For this case, we used $k = 10^{-4}\ m/s$.

station. More snow is available on the flat Plateau compared to the 40/50°-steep slopes, and regular avalanche detachments for safety reasons further reduce snow availability. A basal ice layer at the cold interface rock-snow is supposed to strongly

limit snowmelt penetration in rock walls (Phillips et al., 2016; Ben-Asher et al., 2023). Still, Kneisel et al. (2014) documented thermal disturbance in the underground as soon as snowmelt started, while Kristensen et al. (2021) and Roth and Blikra (2009) recorded large increases in rockslide displacements late in the snowmelting season. Our direct flow measurements show that snowmelt infiltrates every year and that the timing of discharge for fracture L1 fits with SP modeling from the Plateau, which lies southward. This confirms that the catchment of L1 is on the southern slope and that a basal ice layer is not blocking

meltwater flow here. In contrast, a temporary basal ice layer or frozen fractures cannot be excluded for L2, given its starting



delay in some years. A seasonal basal ice layer could also be present on the steep northern slopes, which are impossible to reach in winter and where flow happens later in the season, as observed during fieldwork.

Summer precipitations reach the tunnel only for events above $\approx$ 30 mm/24 h. Poulain et al. (2018) confirm that a saturation threshold is necessary to allow vadose connectivity, although values depend from site to site (Sweetenham et al., 2017). High-intensity short-duration events like thunderstorms often happen at the beginning of precipitation and mainly generate no-flow or only baseflow. Most water does not infiltrate, and instead, it drains away as surface flow due to the total absence of a substrate that acts as a buffer. For the same reason, evaporation effects can be neglected at this location. The amount of precipitation in the period before the event influences the travel time of water in bedrock because pre-saturated fractures have a higher hydraulic conductivity $K$. Zhou et al. (2006) found that infiltration rates and fracture conductivity correlate with fracture patterns at the inlet. Therefore, detecting the precise inlet location could further improve the understanding of the hydrologic system. Unfortunately, the ridge is very exposed, and access is possible only at high risk.

## 6.2 Outflow as the summative system output

The response hydrographs of the two fractures are different: L2 presents a symmetrical hydrograph with similar rising and falling limbs (Fig. 7d to f), while for L1 the rising limb is very steep but the falling limb has a slow decay (Fig. 7e to g). The differences could be due to different fracture filling, catchment shapes, isolated karst areas, or higher connectivity. Fracture density also strongly influence the flow at depth: scarcely fractured networks have a slightly faster response to precipitation than a denser network (Sweetenham et al., 2017). The total annual outflow for L1 reached the decennial maximum value in 2019 with 8500 l/year, which means a theoretical average of 23 l/d. That year, one single snowmelt period recorded a total of 2300 l in 3 days, 27% of the total, showing that extreme events dominate water dynamics here. Measured maximum flow rates are 4 to 10 times higher than those computed by SP and measured by Rist and Phillips (2005) because of a bigger measurement depth and of an increased connectivity of the fracture system during high discharges (Sweetenham et al., 2017). Maximum discharges from snowmelt were reached at the end of the melting season, corresponding to June at this location. For the same period, Weber et al. (2017), Etzelmüller et al. (2022), and Leinauer et al. (2024) demonstrated a clear correlation between snowmelt and increased displacement of unstable slopes. One single rainfall event produced extreme values similar to snowmelt but for a shorter time. A contribution to flow from the neighbor permafrost bodies thaw is theoretically possible. The amounts would be small and visible in dry periods at the end of summer, but it is hard to find signs of it in our data.

## 6.3 Pressurised water inside fractures as an agent driving slope instabilities

The computed hydrostatic pressures strongly depend on the selected parameters, although according to our premises, only limited pairs of hydraulic conductivity $K$ and effective length $L$ are reasonable. The presented values are computed for $K = 10^{-4}$ m/s and $L$ = 11.5-12-12.5 m, which are the most likely parameters, but the same results could be obtained using, for example, $K = 5 \cdot 10^{-5}$ m/s and $L$ = 5.75-6-6.25 m. Still, other $K$-$L$ couples could lead to different hydraulic pressures. Much higher hydraulic conductivities are measured on the Zugspitzplatt by Rappl et al. (2010) with tracers, but this is due to the well-developed karst system present at that location, which is not present on the slope.



In extreme cases, hydrostatic pressures up to 40 m $\pm$ 10 m can be reached, similarly to the models of Magnin and Josnin
(2021). Average values reach 20 m during summer rainfall events and 27 m during spring snowmelt, pressures that can be mechanically critical (Scandroglio et al., 2021). These levels are achieved many times in the summer season and every day during snowmelt, generating repeated loading-unloading cycles that have been rarely considered but can be a crucial destabilizing factor for slope instabilities. Leinauer et al. (2024) and Helmstetter and Garambois (2010) reported that every drop of water can accelerate or trigger instabilities. This can be true only for superficial movements since our no-flow measurements and the
models of Sweetenham et al. (2017) show that minor events do not reach depths of 25-50 m.

Out model provides qualitative estimates of fracture's saturation level at depth, which is crucial to understand permafrost evolution and rock wall destabilization patterns (Magnin and Josnin, 2021). When fractures are fully saturated, we must also consider other destabilizing effects like the reduction of cohesion and friction of the fracture's filling material and the reduction of shear strength by counteracting the normal stresses (Scandroglio et al., 2021). During snowmelt, water flows uninterrupted
for many weeks, and fractures remain longer saturated, while in summer, rain events alternate dry periods, so fracture saturation is highly variable, and destabilizing effects last shortly.

## 6.4 Error sources and uncertainties

**Input:** Due to the elevation of the study site and our classification, the number of snow events (N>100) is much bigger than the number of rain events (N=23), which influences statistics quality. Snowpack modeling is computed only in 1D at a different
elevation than the ridge: 2D or 3D modeling could improve the fitting of the melting phase. Rainfall events are analyzed only hourly, while a 10-minute resolution would provide better insight into high-intensity, short-duration events. Due to their nature, extreme events are rare and statistically less represented.

**Outputs:** Both loggers suffered repeated failures due to lightning strikes, battery problems, and maintenance. Therefore, data gaps could be mistaken for no-flow events or for ice sealing the fracture. L1 and L2 don't always behave similarly, e.g.,
discharge hydrographs differ, but this analysis focuses mostly on L1 because only a few events were recorded by L2. In fact, the latter is more prone to failure and shows variations in peak discharge with time that are not clearly explainable.

**System:** Calibration took place using one extreme rain event that could present higher $K$ and higher fracture interconnectivity than normal events. To include this variability, all events are incorporated in the recession curve that connects discharge to event length. We have also chosen robust estimates and performed sensitivity propagation to check the robustness of the results.
Bedrock deeper than 8 meters is expected to be less densely fractured (Clarke and Burbank, 2011), and a conductivity reduction up to 65% is expected close to a tunnel due to stress increase and joint closure (Fernandez and Moon, 2010). These effects can strongly influence pressure (Montgomery et al., 2002). $K$ varies according to fracture pre-saturation (Fig. 8c); porosity and hydraulic conductivity are, very likely, not uniform in space, but for simplicity, we don't include these variations in our model. Due to the increase of interconnectivity for high discharges, we cannot exclude that in extreme events, water spreads laterally,
which produces hydraulic heads smaller than computed.



## 6.5 Future changes in the system

Snowmelt is expected to begin up to one month earlier by the end of this century (Vorkauf et al., 2021). If so, melting rates will be slower due to the reduced solar radiation early in the year (Musselman et al., 2017), leading to a partial reduction of snowmelt infiltration rates and so lower hydrostatic pressures in fractures. Snow-free periods might increase and so the number of summer rainfall events. Heavy precipitation will generally become more frequent and more intense with global warming (IPCC, 2023). Accordingly, we expect i) more events with liquid precipitation and ii) more frequent and more intense extreme flow events in fractures, resulting in higher hydrostatic pressures. Due to climate change, the recently enhanced permafrost degradation increases active layer thickness, creates new horizontal and vertical flowing paths, and affects massively fracture permeability. In fact, unfrozen fractures are up to three magnitudes more permeable than frozen ones (Pogrebiskiy and Chernyshev, 1977), with significant effects on hydrostatic pressure. Pressurized water in fractures boosts permafrost degradation and could become more important than thermal propagation.

## 7 Conclusions

This study combines a decade of meteorological data, snowmelt modeling, and discharge measurements, and thereby provides novel insights into water dynamics in degraded permafrost bedrock. We describe in detail the timing and the quantities of water infiltration in deep fractures after snowmelt and rainfall events and estimate the possible resulting hydrostatic pressures.

At this elevation, snowmelt produces, on average and in total, higher discharges than rainfall events, while extreme events are similar in both cases and can reach up to 800 l/d. Due to climate change, this dynamic might shift towards more rain and less snowmelt by the end of the century. Rainfall reaches the 55m-deep tunnel with an average delay of 31 h, but this value decreases when the fractures are pre-saturated, e.g., during snowmelt periods. No-flow and baseflow events are indicators of unsaturated and partially saturated fractures, respectively. Fully saturated fractures require > 30 mm precipitation within 24 h, but high-intensity short-duration rain barely contributes to fracture flow. The discharge curves of summer precipitation fit into a general recession curve composed of two exponential terms for laminar flow, which allows forecasting the duration of an event, given its discharge. With the help of a simple empirical fracture flow model, we detected flow anomalies that can be explained by saturation changes and water storage. The fracture can store up to 550 l in extreme events, which is expected to fully saturate the fracture and increase its interconnectivity. The hydraulic head resulting from the water accumulation is computed using Darcy's Law for a falling head together with the recession curve. On a daily mean, hydrostatic pressures can reach 27 m ± 6 m during snowmelt, while rain events generate slightly lower pressures. Snowmelt generates long-lasting pressures for weeks, with daily cycles that can strongly reduce slope stability. Extreme events produce discharges up to 58 l/h in the tunnel, resulting in hydrostatic pressures of 40 m ± 10 m (400 kPa). These values are definitely enough to trigger unstable slopes.

Here we quantitatively demonstrate the relevance of water flow in deep fractures and prove its relevance for slope stability of degraded bedrock permafrost. The estimated hydrostatic pressures can destabilize and/or trigger unstable rock slopes. The combination of climate change and hydrostatic pressures in periglacial areas amplifies permafrost degradation so that in the near future, water is expected to reach new paths and deeper levels, producing higher pressures, thus increasing the hazard.



*Code and data availability.* Discharge data, modeled snowmelt, precipitation data, and the corresponding codes for data analysis will be published on Zenodo before the publication of this manuscript (**?**). Weather data for snow modeling can be obtained from the Bavarian Avalanche Warning Service (Lawinenwarndienst im Bayerischen Landesamt für Umwelt). Weather data from the summit can be obtained from the DWD German Weather Service at https://cdc.dwd.de/portal/.

*Author contributions.* RS designed the study under the supervision of MK. RS conducted the fieldwork with the support of TR. RS and SW developed the concept of the study, the data analysis, and the model. RS performed the analysis, implemented the model, and made the figures in Matlab. RS prepared the manuscript with revision and final approval from all authors.

*Competing interests.* At least one of the (co-)authors is a member of the editorial board of Earth Surface Dynamics. The authors have no other competing interests to declare.

*Acknowledgements.* This study was supported by the AlpSenseRely project, funded by the Bavarian State Ministry of the Environment and Consumer Protection (TUS01UFS-76976), and by the Hydro-PF project, funded by the TUM International Graduate School of Science and Technology IGSSE (Team 12.9). Special thanks to the Environmental Research Station Schneefernerhaus and the Bayerische Zugspitzbahn Bergbahn AG for the amazing logistic support. We thank all the students and colleagues who provided support in more than 100 days of fieldwork. We are also grateful to Franziska and Thomas from the Bavarian Avalanche Center for the snow data.



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
