# Peer review of "Decadal in-situ hydrological observations and empirical modeling of pressure head in a high-alpine, fractured calcareous rock slope"

_EGUsphere, 2024_

## Referee Comment (RC1)

Review of *'An empirically-derived hydraulic head model controlling water storage and outflow over a decade in degraded permafrost rock slopes (Zugspitze, D/A)'* for **ESurf** by Scandroglio et al.
Luc Illien

**1. Overview and suggestion**

The paper constrains the contribution of snow melt relative to precipitation events for feeding a fracture aquifer that can be monitored thanks to a unique set-up, which measures discharge at two fracture outputs in a tunnel located under the Zugspitze, the highest peak in Germany. With the measurements, they quantify the groundwater storage in the fracture network. Additionally, they propose an empirical model based on the Darcy's equation for constraining the hydraulic head above the tunnel.

The paper is highly relevant: Mountain groundwater is a current hot topic in Hydrological studies and ground observations are needed to calibrate larger scale models and understand how much water mountains can store, a crucial question for mitigating freshwater resources. Unfortunately, mountain groundwater observations are rather scarce: This study fills this observational gap in a unique high mountain environment. However, in its current form, the paper is not clear about its aim and findings. Not that there are none but rather because of the writing and presentation. I have doubts that the paper was extensively checked and reviewed by the co-authors before submitting. Additionally, I have one question for the authors: Is it a hydrology paper or a geomorphology one ? I struggle to understand why it was addressed to Earth Surface Dynamics and not HESS. In brief: The science is good but the presentation makes it challenging to digest.

Nevertheless, because the results are highly relevant for the mountain groundwater community and re-analysis is not necessary, I would still recommend **a major revision** for this paper.

**2. General Comments**

A. The title is misleading to me: there is at the moment a high emphasis on the empirical model where most of the paper is about reporting ( very interesting) field observations. The model just serves as an interpretation and a starting point for the discussion in my opinion. Also why being so specific ? The 'permafrost slopes' are barely addressed and I don't feel this is the object of the study. Something like **In-situ observations of the dynamics of an alpine mountainous fractured aquifer** would be more general and accurate (just an example). Sorry to be annoying but this is for your readership and it is important ! Also: What is D/A ? (Yes I know it's Deutschland/Austria but it may not connect with everybody) .

B. The structure of the paper is not clear and makes the reading difficult. The authors introduce a lengthy method section, explaining also their model at the same time. Then, they proceed to show the results section, in which they

show the model results at the end of the section. I would advise to make a separate section regarding the model so that you have: Methods → Results (This is what informs and motivates the model you build here and the point of the paper !) → Empirical Model → Discussion. This would be straying a bit from the usual paper structure but you would gain clarity.

C. I cannot find anywhere a clear key message for the study. What's your new finding (Snowmelt vs Rain), a quantitative estimate (what is the storage in the fracture, or a relevant timescale ?) etc… Make it clear in your conclusion and at the end of the intro. I think it would be worth emphasizing how much relative storage is actually entering the fracture. It is great to have such a measure at a 50m depth.

D. The knowledge gap is not well defined in the intro as well. Are you talking about the water budget in the mountains ? Landsliding ? I struggle to see the typical introduction architecture**:** *Motivation -> Knowledge Gap -> The solution you bring and the question you want to address (not the current 4 listed!) -> A brief overview of the findings.*

E. I would keep the statement about slope stability in the discussion and remove it from the introduction for instance. Make it a full hydrological paper and commit to the choice. Some of the cited literature does not advance the paper.

F. The paper is full of words, abbreviations and phrasing that lengthens or distract from the reading in my opinion. For example at the start of the discussion where the sentence gives twice the same info: *'Differently from previous studies, here, for the first time'.*
The paper could benefit from a trim of at least one third of the current version: some sentences are too long and unclear and some paragraphs do not add to the context of the study. I started to note all grammar and organisational mistakes (not reported in this review !) but I stopped as it is not the job of the reviewer and is very time-demanding. I suggest checking throughout before sending it again to reviewers.

**3. Specific comments**

**Abstract:**
I would clean the abstract. Examples:
1→ you start with a permafrost sentence… The paper is not about permafrost induced slope instability. It's about the water budget… Find another opening.
6→ data set of meteorological data → meteorological data. Check throughout.
10→ 'We developed', who is we ? Did you develop a recession curve ? Not the right verb. Check throughout.
15 → 'Here, we show' sounds like the start of a method, So that would be higher before summing up your results. This cuts the flow of the abstract.

**Intro:**
The third paragraph arrives out of the blue after the first two on hydrology. Does this paragraph contribute to your paper ?
Line 55 to 60 → Would fit better in the discussion part of the paper.

**Study Site description and characterisation:**
Line 106 → How do your results compare to the gravimetric measurements if you mention it ?
Line 117 → 'The mapping agrees only partially'. If you write it, I just want to ask why ?

**Methods and Data:**
Line 121 → This pre method paragraph could be in the intro.
Line 140 → I think it is a lot of abbreviation to grasp. Also → Why calling your precipitation  PSUM and not P ?
Line 174 → Title is unclear. What about Modelling of … fluid flow etc..
Line 156 → 'are united' → 'are combined' check throughout (line 160 also)
Line 180 → I am not sure we need this info.
Line 177 → 'a likelihood' → this word is statistically connotated. Are you sure this is the one you want ?

- I found that the part on the recession is too long (from 188 to 195).
- Introduce your variable after the equations, and make the equation part of your sentences
- The time dependency '(t)' is often omitted in almost all equations, please add it or say that you don't mark it for brevity.

Line 211 → 'would also strongly influence the flow behaviour…'
 How can you make this statement at this point of the paper ? We haven't yet seen your data.

Line 215 → I would have this part in a separate modelling section as stated above.
Lines 234 → 'yq' typo ?

**A note on the model:** Would it be possible to list all the assumptions you are making ? For instance, Lateral flow in the fracture network is not consider for your Darcy Head where you assume a main conduit ?

**Results and data interpretation:**

Line 261 → 'earlier than the model' → Which model ?
Line 308 → When you use 'bigger' to describe a variable, I think it would me more correct to say 'larger'. Check throughout.
Part 5.4 → There's a lot of text and data from the recession analysis. Yet, no recession curves are shown in the main text. I think you should show it.
Line 355 → Why thes values for L ? After all, you have 50m of rocks above the tunnel ?

**Figure 1**: The little table is confusing: It looks like the measurements are about the loggers and not the fractures.

**Figure 2:** It's a nice figure but Would it be possible to have fewer abbreviations ? As it is, you need to do mental gymnastics to recall all the abbreviations.

**Figure 3:** Nice figure but it looks the water is in the void of the fracture. It would be nice to indicate, you consider the fracture to have finer materials in which the water circulates. Also what you call 'baseflow' → It doesn't seem to correspond to the classic hydrological definition. Could you define what you consider baseflow in the paper ?

**Figure 7:** Would you consider having somewhere a plot of the Precipitation vs the Flow of the fracture ? That would show the transient storage in the fracture in a nice concise way.

---

## Referee Comment (RC2)

**Review**

An-empirically-derived hydraulic head model controlling water storage and out flow over a decade in degraded permafrost rock slopes (Zugspitze, D/A)

By Riccardo Scandroglio et al.

**General Comments:**

This paper evaluates the ground water dynamics in two rock fractures in a tunnel by deriving an empirical model using decades of discharge and weather measurements and snow simulations. The authors analyzed and compared long terms and short terms trends by combining data records at over three stations in the study region. The authors used water collecting systems to estimate discharges, which were later used to generate fluid flow models beneath the study region. The results are summarized in **7. Conclusion:**

*Here we quantitatively demonstrate the relevance of water flow in deep fractures and prove its relevance for slope stability of degraded bedrock permafrost. The estimated hydrostatic pressures can destabilize and/or trigger unstable rock slopes.*

The technical quality of the paper is good although there are considerable grammar/writing mistakes that should be corrected before publishing. Authors have clearly analyzed and interpreted the results. The novelty of the research is also acceptable. However, I have some suggestions for authors to improve the quality of the manuscript before publishing it. Therefore, I recommend accepting this article with major revisions. My suggestions are as below:

The authors assembled a large data set and rigorously processed it to generate models and get meaningful findings. However, it is somewhat disappointing that the presentation and writing are on a low scale. It is not clear to me why you did this study, what are key findings and how those results will enhance our current understanding of the problems. Please clearly summarize key questions, findings and major impact in both abstract and conclusion sections. There is lot of information and repetitions on intro and method sections that you do not need to interpret your results. You can remove those and that will help to reduce the length of the method section. I would suggest re-structuring the manuscript in the following way: (1) intro (2) Geological or hydrological background and previous work in the region (3) method (4) results including modeling part (5) discussion (6) conclusion. This will follow the general structure that most of the journals follow, and it will be easy to go over the info on the manuscript for both authors and readers.

**Specific comments:**

Title: What is D/A?

1. Line 8: We analyze input (i.e., snowmelt …) and outputs (i.e.,…, base flow(partially saturated), no-flow(unsaturated)).

2. Line 14: Remove sentence starting "E.g.,".

3. Line 15: Sentence starts on "Here we show", Not clear you put this on right place. I feel like you are referring to the method within the few sentences that you summarized your results. Either Re-write the sentence or put it into the place where you refer your methodology in the abstract.

4. Line 24-26: Sentence starts on "Logistically challenging terrain ….", Not clear what is your point in this sentence. Please re-phrase it.

5. Line 30: "Developments suggest that", Not clear, what development? Bedrock or basin or hillslopes or what?

6. Line 53: Remove "of rock-rock contacts" and add "between rock contacts"

7. Line 61-65: As I understand, authors try to list the methods and their limitations in this paragraph, but it is not clearly state that. So please re-write the entire paragraph.

8. Line 78: remove "and visited by thousands of tourists daily"
9. Line 79: remove all sentence starting " A disused…"

10. Line 101: Replace "thanks to a" with ", equipped with". I noted similar types of wording at many places (e.g., line 154, 175) in the manuscript and I would recommend either removing or fixing those places with appropriate scientific words. You can give credits to all in the acknowledgement section.

11. Line 120: title should be like "4. Data and method"

12. Line 125: Add "snowmelt prediction or calculated snowmelt"

13. Line 136: Add citation for software package.

14. Line 234: replace "yq" with "q"

Figures:
Fig 1: add description about little inset map on figure (a) to caption. shift small table beneath figure (d).

Fig 6: fig (d & g-i) and Fig 8a: Not sure why there are color lines outside from the axis and what is referring to? Never seen manuscript figure like that before.

---

## Author Comment (AC1)

Dear Editor,

We are pleased to submit the revised manuscript entitled *"Decadal in-situ hydrological observations and empirical modeling of pressure head in a high-alpine, fractured calcareous rock slope"* (egusphere-2024-1512).

We thank both referees for the constructive feedback and input on the previous submission. We addressed all the issues raised and believe that the implemented changes have substantially improved the revised manuscript. We briefly outline the primary changes here and add a one-by-one reply (in blue) to the reviews on the following pages.

- As suggested by both reviewers, we worked a lot on the clarity of the text, improving the language, avoiding repetitions, and reducing the text when possible. This included major revisions of the complete text.
- According to the comments, the structure of the paper has been changed. We moved the the model part (now Chapter 5: recession curve, anomalies detection, storage model and hydrostatic pressures) after the hydrological analysis (Chapter 4).
- We also added a graphical abstract to help readers understand the article's content.

We sincerely hope that these revisions and explanations address all of your concerns and improve our manuscript's quality and clarity.

With kind regards,
Riccardo Scandroglio
On behalf of all authors
* * *
REVIEWER 1

1a) Is it a hydrology paper or a geomorphology one? I struggle to understand why it was addressed to Earth Surface Dynamics and not HESS.

> Although the suggestion of addressing the paper to HESS is good, we think it also well fits to ESD, according to the following definition of the journal: "*ESD solicits contributions that investigate [interactions among various Earth systems] and their underlying mechanisms, ways how these can be conceptualized, modeled, and quantified, predictions of the overall system behavior to global changes, and the impacts for its habitability, humanity, and the future functioning of the Earth system in the Anthropocene.*"
>
> Our research questions arise from the geomorphological interrogative: Can water destabilize a rock slope? Of course, we are investigating a hydrological topic, nevertheless from the perspective of natural hazards researchers. Nevertheless, the collected data offer a worthy dataset for hydrologists to better describe water dynamics at depth.
>
> Therefore, we also implemented the chapter structure as suggested by the reviewers and now propose two distinct chapters that represent the two focuses:
> - Chapter 4, "Results and interpretation", focuses on the innovative dataset for deep groundwater dynamics in bedrock and is precious for the alpine hydrology community.
> - Chapter 5, "Model fluid flow and water accumulation…", is relevant for geomorphologists and modelers in the field of natural hazards in order to evaluate hydrostatic pressures and their implications for slope stability.

2a) The title is misleading to me: there is at the moment a high emphasis on the empirical model where most of the paper is about reporting (very interesting) field observations. The model just serves as an interpretation and a starting point for the discussion in my opinion. Also why being so specific? The 'permafrost slopes' are barely addressed and I don't feel this is the object of the study. Something like In-situ observations of the dynamics of an alpine mountainous fractured aquifer would be more general and accurate (just an example). Sorry to be annoying but this is for your readership and it is important! Also: What is D/A ? (Yes I know it's Deutschland/Austria but it may not connect with everybody) .

> Thanks for the good point and for the constructive suggestions. We agreed and changed the title, emphasizing the in-situ observations and removing the unnecessary references to permafrost (not the topic here).
> The new title is: "*Decadal in-situ hydrological observations and empirical modeling of pressure head in a high-alpine, fractured calcareous rock slope*".

3a) The structure of the paper is not clear and makes the reading difficult. The authors introduce a lengthy method section, explaining also their model at the same time. Then, they proceed to show the results section, in which they show the model results at the end of the section. I would advise to make a separate section regarding the model so that you have: Methods → Results (This is what informs and motivates the model you build here and the point of the paper!) → Empirical Model → Discussion. This would be straying a bit from the usual paper structure but you would gain clarity.

> As suggested by both reviewers, we updated the structure of the paper, shifting the model to a separate section after the hydrological results. The new structure is as follows:
> Chapter 4) "Results and interpretation" includes:
>
> - Snowmelt induced discharge
> - Rainfall induced discharge
> - No-flow and extreme events
>
> Chapter 5) "Model fluid flow and water accumulation in deep-bedrock fractures" includes:
>
> - Recession-curve fitting
> - Flow anomalies detection
> - Fracture saturation and storage model
> - Estimating hydrostatic pressures from discharge

4a) I cannot find anywhere a clear key message for the study. What's your new finding (Snowmelt vs Rain), a quantitative estimate (what is the storage in the fracture, or a relevant timescale?) etc… Make it clear in your conclusion and at the end of the intro. I think it would be worth emphasizing how much relative storage is actually entering the fracture. It is great to have such a measure at a 50m depth.

5a) The knowledge gap is not well defined in the intro as well. Are you talking about the water budget in the mountains? Landsliding? I struggle to see the typical introduction architecture: Motivation -> Knowledge Gap -> The solution you bring and the question you want to address (not the current 4 listed!) -> A brief overview of the findings.

> Thanks for the two comments. We fully reviewed and strongly improved the abstract, the introduction, and the conclusions, clarifying the research gap and the two focuses of this

article. The research gap and the research questions also have a new formulation, to improve clarity.

Still, the results on water dynamics are a lot and cover a wide range, therefore, it's hard to resume them in a few lines. In the conclusions, we list what we think is more important, but according to the focus of the reader, the interest can be on other information (someone can be more interested in the delay times, others in the quantities).

6a) I would keep the statement about slope stability in the discussion and remove it from the introduction for instance. Make it a full hydrological paper and commit to the choice. Some of the cited literature does not advance the paper.

We thank the reviewer for the feedback. Considering our answer 1a) of this letter, we try to keep the balance between two focuses, the hydrological and the geomorphological. Therefore, we find it necessary to introduce the slope stability in the introduction.

7a) The paper is full of words, abbreviations and phrasing that lengthens or distract from the reading in my opinion. For example, at the start of the discussion where the sentence gives twice the same info: 'Differently from previous studies, here, for the first time'. The paper could benefit from a trim of at least one third of the current version: some sentences are too long and unclear and some paragraphs do not add to the context of the study. I started to note all grammar and organisational mistakes (not reported in this review !) but I stopped as it is not the job of the reviewer and is very time-demanding. I suggest checking throughout before sending it again to reviewers.

We improved the language thoroughly, shortened sentences, enhanced clarity, and removed repetitions. Almost 2 pages have shortened the paper, and we think further trimming would make the content not clear.
* * *
8a) Specific comments

Abstract: I would clean the abstract. Examples: 1→ you start with a permafrost sentence... The paper is not about permafrost induced slope instability. It's about the water budget... Find another opening.

Thanks for the comment; the abstract has been cleaned and rephrased, especially the opening.

L6→ data set of meteorological data → meteorological data. Check throughout. - Changed.

L10→ 'We developed', who is we? Did you develop a recession curve? Not the right verb. Check throughout. - Changed.

L15 → 'Here, we show' sounds like the start of a method, So that would be higher before summing up your results. This cuts the flow of the abstract. - Rephrased.

Intro: The third paragraph arrives out of the blue after the first two on hydrology. Does this paragraph contribute to your paper?

We think this is an important contribution to the paper, but agree that it was unclear. We rephrased and shortened it, clarifying why it is important here.

L55 to 60 → Would fit better in the discussion part of the paper.

According to answers 1a) and 6a) of this letter, we moved some parts of this paragraph to the discussion and rephrased the relevant parts for the introduction.

Study Site description and characterization: Line 106 → How do your results compare to the gravimetric measurements if you mention it?

The results from the gravimetric measurements published so far are only at the catchment scale. They could well reproduce the LWC of the snow at the catchment scale. Experiments at smaller scales are being conducted.

Line 117 → 'The mapping agrees only partially'. If you write it, I just want to ask why?

We added the reason: "This difference is mainly due to the influence of the fault zone."

Methods and Data: Line 121 → This pre method paragraph could be in the intro. - Rephrased.

Line 140 → I think it is a lot of abbreviation to grasp. Also → Why calling your precipitation PSUM and not P?

Thanks for the constructive comment; we changed precipitation to P and tried to reduce abbreviations, e.g., in the new Figure 8.

Line 174 → Title is unclear. What about Modelling of … fluid flow etc..

Title modified into: "Modelling flow and storage in deep-bedrock fractures"

Line 156 → 'are united' → 'are combined' check throughout (line 160 also) - Suggestion accepted.

Line 180 → I am not sure we need this info. - Removed

Line 177 → 'a likelihood' - this word is statistically connotated. Are you sure this is the one you want?

Rephrased: *"Quantities cannot be considered as precise as measures".*

- I found that the part on the recession is too long (from 188 to 195).

This part has been reduced.

- Introduce your variable after the equations, and make the equation part of your sentences

Done. We prefer to leave the equation separated for clarity.

The time dependency '(t)' is often omitted in almost all equations, please add it or say that you don't mark it for brevity.

We added time dependency in section 5.3.

Line 211 → 'would also strongly influence the flow behaviour…' How can you make this statement at this point of the paper? We haven't yet seen your data.

We rephrased this sentence:
*"Results of the recession curve confirm that the presence is possible but only to a limited extent. Extensive karst voids are present under the Plateau (Wetzel, 2004), but their presence in our study area has not been proven yet. Therefore, we exclude this possibility."*

Line 215 → I would have this part in a separate modelling section as stated above. - Done

Lines 234 → 'yq' typo? - Yes, corrected

A note on the model: Would it be possible to list all the assumptions you are making? For instance, Lateral flow in the fracture network is not consider for your Darcy Head where you assume a main conduit?

> We listed all assumptions in chapter 6.4., including "Lateral flow is neglected."

Results and data interpretation: Line 261 → 'earlier than the model' → Which model?

> Snowpack modeling – has been added for clarity.

Line 308 → When you use 'bigger' to describe a variable, I think it would me more correct to say 'larger'. Check throughout.

> "Bigger" was substituted by "larger" throughout the paper.

Part 5.4 → There's a lot of text and data from the recession analysis. Yet, no recession curves are shown in the main text. I think you should show it.

> We moved the figure about recession analysis from supporting material to the main text (Fig.7) and exchanged it with the error table, which is now in the supporting material (Tab.S1)

Line 355 → Why these values for L? After all, you have 50m of rocks above the tunnel?

> L is defined as the "effective path" that offers resistance to water, i.e. the "breaking" path. We assume this to be only a limited part of the fracture, smaller than the hydraulic head, as we think this case is more realistic. Other options are also possible but not considered here.

Figure 1: The little table is confusing: It looks like the measurements are about the loggers and not the fractures.

> We changed the column's name from "Logger" to "Fracture" and added an arrow to clarify. A drawing of the fracture with dip is also included.

Figure 2: It's a nice figure but would it be possible to have fewer abbreviations? As it is, you need to do mental gymnastics to recall all the abbreviations.

> Thanks for the suggestions; abbreviations have been removed.

Figure 3: Nice figure but it looks the water is in the void of the fracture. It would be nice to indicate, you consider the fracture to have finer materials in which the water circulates. Also what you call 'baseflow' → It doesn't seem to correspond to the classic hydrological definition. Could you define what you consider baseflow in the paper?

> Thanks for the suggestion. We improved the image by adding colors and description.

> You are right; the word "baseflow" is not 100% the same as the classical hydrology definition, but they are still very close. Our definition can be found in Section 3.3, Line 129.

Figure 7: Would you consider having some plot of the Precipitation vs the Flow of the fracture? That would show the transient storage in the fracture in a nice concise way

> It is unclear to us what "Precipitation vs Flow" means: a scatter plot? Or plot of differences? A plot of differences is present in Figure S13 of the Supplementary Material to precisely show the transient storage for all events.

REVIEWER 2

1b) The technical quality of the paper is good although there are considerable grammar/writing mistakes that should be corrected before publishing. (…) The authors assembled a large data set and rigorously processed it to generate models and get meaningful findings. However, it is somewhat disappointing that the presentation and writing are on a low scale.
It is not clear to me why you did this study, what are key findings and how those results will enhance our current understanding of the problems. Please clearly summarize key questions, findings and major impact in both abstract and conclusion sections.

> Thank you for the feedback; we reviewed the presentation and writing overall to improve the quality of the paper. We formulated the abstract and the conclusions so that research gap (L1-6), key questions (L6-7), findings (L10-15) and major impact (L15-17) are more clear.

2b) There is lot of information and repetitions on intro and method sections that you do not need to interpret your results. You can remove those and that will help to reduce the length of the method section. - See answer 7a

I would suggest re-structuring the manuscript in the following way: (1) intro (2) Geological or hydrological background and previous work in the region (3) method (4) results including modeling part (5) discussion (6) conclusion. This will follow the general structure that most of the journals follow, and it will be easy to go over the info on the manuscript for both authors and readers.

> Thanks for the feedback, we adapted the structure of the paper. See answer 3a) for details.
* * *
Specific comments: Title: What is D/A?

> Deutschland/Austria -> this has been removed and the title changed according to the requirements of the first reviewer.

Line 8: We analyze input (i.e., snowmelt …) and outputs (i.e.,…, base flow(partially saturated), no-flow(unsaturated)). – Thanks, we corrected it.

Line 14: Remove sentence starting "E.g.,". - Thanks, we removed it.

Line 15: Sentence starts on "Here we show", Not clear you put this on right place. I feel like you are referring to the method within the few sentences that you summarized your results. Either Re-write the sentence or put it into the place where you refer your methodology in the abstract.

> Thank you for the suggestion, this sentence and most of the abstract have been rephrased.

Line 24-26: Sentence starts on "Logistically challenging terrain ….", Not clear what is your point in this sentence. Please re-phrase it. - Thanks, the sentence has been rephrased.

Line 30: "Developments suggest that", Not clear, what development? Bedrock or basin or hillslopes or what? - Here, it means research development. We reformulated the sentence for more clarity.

Line 53: Remove "of rock-rock contacts" and add "between rock contacts"

> This vocabulary is taken from the cited study (Krautblatter et al 2013), but we accept the comment and changed the sentence, in order to improve understanding for the reader.

Line 61-65: As I understand, authors try to list the methods and their limitations in this paragraph, but it is not clearly state that. So please re-write the entire paragraph. - Thanks, rephrased.

Line 78: remove "and visited by thousands of tourists daily" - Removed.

Line 79: remove all sentence starting " A disused…"

> We understand the reason of the comment, but we think this sentence is important for understanding the study site. Therefore, we rephrased and shortened the sentence.

Line 101: Replace "thanks to a" with ", equipped with". - Rephrased.

I noted similar types of wording at many places (e.g., line 154, 175) in the manuscript and I would recommend either removing or fixing those places with appropriate scientific words. You can give credits to all in the acknowledgement section.

> All the similar wording has been substituted with scientific words.

Line 120: title should be like "4. Data and method" - Changed to "Data and methods".

Line 125: Add "snowmelt prediction or calculated snowmelt" - Rephrased.

Line 136: Add citation for software package.

> Added: Lehning, M., Bartelt, P., Brown, B., Russi, T., Stöckli, U., and Zimmerli, M.: SNOWPACK model calculations for avalanche warning based upon a new network of weather and snow stations, Cold Regions Science and Technology, 30, 145–157, https://doi.org/10.1016/S0165-232X(99)00022-1, 1999.

Line 234: replace "yq" with "q" - Done

Fig 1: add description about little inset map on figure (a) to caption. Shift table beneath figure (d)

> Thanks for the suggestions, both have been implemented.

Fig 6: fig (d & g-i) and Fig 8a: Not sure why there are color lines outside from the axis and what is referring to? Never seen manuscript figure like that before.

> These graphs are scatter plots with marginal histograms, produced with the function "*scatterhistogram*" in Matlab. In this case, kernel density has been used to produce the lines (marginal histograms). This extra information helps the reader localize values with the highest concentration at a glance.